# Cortical tracking of hierarchical rhythms orchestrates the multisensory processing of biological motion

Li Shen[1,2], Shuo Li[1,2], Yuhao Tian[1,2], Ying Wang[1,2]*, Yi Jiang[1,2]

[1]State Key Laboratory of Cognitive Science and Mental Health, Institute of Psychology, Chinese Academy of Sciences, Beijing, China; [2]Department of Psychology, University of Chinese Academy of Sciences, Beijing, China

## eLife Assessment

Wang et al. presented visual (dot) motion and/or the sound of a walking person and found **solid** evidence that EEG activity tracks the step rhythm, as well as the gait (2-step cycle) rhythm, with some demonstration that the gait rhythm is tracked superadditively (power for A+V condition is higher than the sum of the A-only and V-only condition). The **valuable** findings will be of wide interest to those examining biological motion perception and oscillatory processes more broadly.

**\*For correspondence:**
wangying@psych.ac.cn

**Abstract** When observing others' behaviors, we continuously integrate their movements with the corresponding sounds to enhance perception and develop adaptive responses. However, how the human brain integrates these complex audiovisual cues based on their natural temporal correspondence remains unclear. Using electroencephalogram (EEG), we demonstrated that rhythmic cortical activity tracked the hierarchical rhythmic structures in audiovisually congruent human walking movements and footstep sounds. Remarkably, the cortical tracking effects exhibit distinct multisensory integration modes at two temporal scales: an additive mode in a lower-order, narrower temporal integration window (step cycle) and a super-additive enhancement in a higher-order, broader temporal window (gait cycle). Furthermore, while neural responses at the lower-order timescale reflect a domain-general audiovisual integration process, cortical tracking at the higher-order timescale is exclusively engaged in the integration of biological motion cues. In addition, only this higher-order, domain-specific cortical tracking effect correlates with individuals' autistic traits, highlighting its potential as a neural marker for autism spectrum disorder. These findings unveil the multifaceted mechanism whereby rhythmic cortical activity supports the multisensory integration of human motion, shedding light on how neural coding of hierarchical temporal structures orchestrates the processing of complex, natural stimuli across multiple timescales.

## Introduction

The perception of biological motion (BM), the movements of living creatures, is a fundamental ability of the human visual system. Extensive evidence shows that humans can readily perceive BM from a visual display depicting just a handful of light dots attached to the head and major joints of a moving person (*Blake and Shiffrar, 2007*). Nevertheless, in real life, BM perception often occurs in multisensory contexts. For instance, one may simultaneously hear footstep sounds while seeing others walking. The integration of these visual and auditory BM cues facilitates the detection, discrimination, and attentional processing of BM (*Mendonça et al., 2011*; *Shen et al., 2023a*; *Thomas and Shiffrar, 2013*; *van der Zwan et al., 2009*). Notably, such benefits are diminished when the visual BM

is deprived of characteristic kinematic cues but not low-level motion attributes (*Brooks et al., 2007*; *Shen et al., 2023a*; *Thomas and Shiffrar, 2010*), and the temporal windows of perceptual audiovisual synchrony are different between BM and non-BM stimuli (*Arrighi et al., 2006*; *Saygin et al., 2008*), highlighting the specificity of audiovisual BM processing. This specificity may relate to the evolutionary significance of BM and its pivotal role in social situations. In particular, integrating multisensory BM cues is foundational for perceiving and attending to other people and developing further social interaction. Such ability is usually compromised in people with social deficits, such as individuals with autism spectrum disorder (ASD) (*Falck-Ytter et al., 2013*; *Feldman et al., 2018*). These findings underline the unique contribution of multisensory BM processing to human perception and social cognition. However, despite the behavioral evidence, the neural coding of audiovisual BM cues and its possible link with individuals' social cognitive capability remains largely unexplored.

An intrinsic property of human movements (such as walking and running) is that they are rhythmic and accompanied by frequency-congruent sounds. The audiovisual integration (AVI) of such rhythmic stimuli may involve a process whereby brain activity aligns with and tracks external rhythms, revealed by increased power or phase coherence of neural oscillations at corresponding frequencies (*Bauer et al., 2020*; *Ding et al., 2016*; *Obleser and Kayser, 2019*). Studies based on simple or discrete stimuli show that temporal congruency in auditory and visual rhythms significantly enhances the cortical tracking of rhythmic stimulations in both modalities (*Covic et al., 2017*; *Keitel and Müller, 2016*; *Nozaradan et al., 2012b*). Unlike these stimuli, BM conveys complex hierarchical rhythmic structures corresponding to integration windows at multiple temporal scales. For example, the human locomotion movement has a narrower integration window consisting of each step (i.e. step cycle) and a broader integration window incorporating the opponent motion of the two feet (i.e. gait cycle). A recent study suggests that neural tracking of these nested kinematic structures contributes to the spatiotemporal integration of visual BM cues in different manners (*Shen et al., 2023b*). However, it remains open whether and how the cortical tracking of hierarchical rhythmic structures underpins the AVI of BM information.

To tackle this issue, we recorded electroencephalogram (EEG) signals from participants who viewed rhythmic point-light walkers or/and listened to the corresponding footstep sounds under visual (V), auditory (A), and audiovisual (AV) conditions in Experiments 1 a and 1b (*Figure 1*). An enhanced cortical tracking effect in the AV condition compared to each unisensory condition will indicate significant multisensory gains. Moreover, we adopted an additive model to classify multisensory integration based on the AV vs. A+V comparison. This model assumes independence between inputs from each sensory modality and distinguishes among sub-additive (AV <A + V), additive (AV = A + V), and super-additive (AV >A + V) response modes (see a review by *Stevenson et al., 2014*). The additive mode represents a linear combination between two modalities. In contrast, the super-additive and sub-additive modes indicate non-linear interaction processing, either with potentiated neural activation to facilitate the perception or detection of near-threshold signals (super-additive) or a deactivation mechanism to minimize the processing of redundant information cross-modally (sub-additive) (*Laurienti et al., 2005*; *Metzger et al., 2020*; *Stanford et al., 2005*; *Wright et al., 2003*). Distinguishing among these integration modes may help elucidate the neural mechanism underlying AVI in specific contexts.

Experiment 2 examined to what extent the AVI effect was specific to the multisensory processing of BM by using non-BM (inverted visual stimuli) as a control. Inversion disrupts the unique, gravity-compatible kinematic features of BM but not the rhythmic signals generated by low-level motion cues (*Ma et al., 2022*; *Shen et al., 2023b*; *Simion et al., 2008*; *Troje and Westhoff, 2006*; *Wang et al., 2022*), thus is expected to interfere with the BM-specific neural processing. Participants perceived visual BM stimuli accompanied by temporally congruent or incongruent BM sounds. Comparing the congruency effect in neural responses between the upright and inverted conditions allowed us to verify whether the AVI of BM involves a mechanism distinct from that underlies the AVI of non-BM. In addition, we further explored the possible linkage between the BM-specific neural tracking effect and observers' autistic traits. Previous behavioral studies found reduced orienting to audiovisually synchronized BM stimuli in ASD (*Falck-Ytter et al., 2018*). Since individuals with varying social cognitive abilities lie on a continuum extending from clinical to nonclinical populations with different levels of autistic traits, we investigated whether cortical tracking of audiovisual BM correlates with individuals' autistic traits, as measured by the Autism-Spectrum Quotient (AQ) (*Baron-Cohen et al., 2001*).

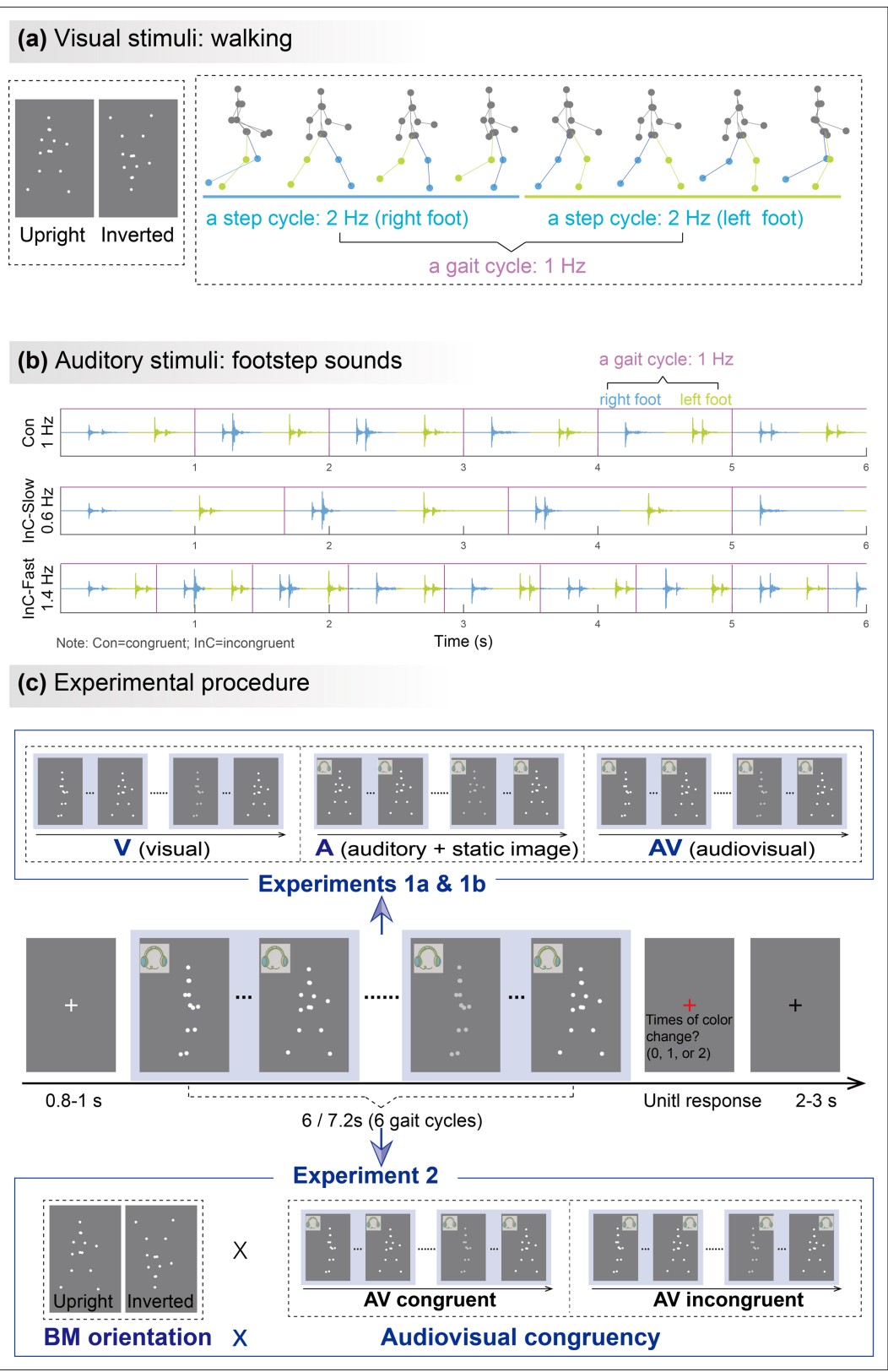

**Figure 1.** Illustrations of audiovisual stimuli and experimental procedures. The illustration was based on stimuli with a gait-cycle frequency of 1 Hz. (**a**) Visual stimuli. The left panel depicts the static schematic of upright and inverted point-light walkers. The right panel shows the keyframes from a gait cycle of the biological motion (BM) sequence. The colors of dots and lines between dots are for illustration only and are not shown in the experiments.

*Figure 1 continued on next page*

*Figure 1 continued*

(**b**) Auditory stimuli. The auditory sequences contain periodic impulses of footstep sounds whose peak amplitudes occur around the points when the foot strikes the ground. The duration of two successive impulses defines the gait cycle of footstep sounds, which is temporally congruent (Con) or incongruent (InC) with the visual stimuli. (**c**) Experimental procedure and design. The color of the visual stimuli changed one or two times within 6 s in the catch trials but did not change in the experimental trials. Participants were required to report the number of changes when the point-light stimulus was replaced by a red fixation. In Experiment 1, participants viewed rhythmic point-light walkers or/and listened to the corresponding footstep sounds under visual (V), auditory (A), and audiovisual (AV) conditions. The visual stimulus was the BM sequence in the V and AV conditions but a static frame from the sequence in the A condition. Experiment 2 included only the AV condition with different stimulus orientations (upright vs. inverted) and audiovisual congruency (congruent vs. incongruent).

## Results

In all experiments, 17–23% of the trials were randomly selected as catch trials, in which the color of the walker changed one or two times throughout the trial, and there was no color change in other trials. Participants were required to detect the color change of visual stimuli (zero to two times during one trial) to maintain attention. Behavioral analysis on all trials showed that their performances for the task were generally high and equally well in all conditions of Experiment 1 a (mean accuracy >98%; $F$ (2, 46)=0.814, p=0.450, $\eta_p^2$=20.034), Experiment 1b (mean accuracy >98%; $F$ (2, 46)=0.615, p=0.545, $\eta_p^2$=20.026), and Experiment 2 (mean accuracy >98%; $F$ (3, 69)=0.493, p=0.688, $\eta_p^2$=20.021), indicating comparable attention state across conditions. The catch trials were excluded from the following EEG analysis.

### Cortical tracking of rhythmic structures in audiovisual BM reveals AVI

### Experiment 1a

In Experiment 1 a, we examined the cortical tracking of rhythmic BM information under V, A, and AV conditions (*Figure 1c*). We were interested in two critical rhythmic structures in the walking motion sequence, i.e., the gait cycle and the step cycle (*Figure 1a and b*). During walking, each step of the left or right foot occurs alternatively to form a step cycle, and the antiphase oscillations of limbs during two steps characterize a gait cycle (*Shen et al., 2023b*). In Experiment 1 a, the frequency of a full gait cycle is 1 Hz, and the step-cycle frequency is 2 Hz. The strength of the cortical tracking effect was quantified by the amplitude peaks emerging from the EEG spectra at these frequencies.

As shown in the grand average amplitude spectra (*Figure 2a*), both the responses in three conditions showed clear peaks at step-cycle frequency (2 Hz; V: $t$ (23)=6.963, p<0.001; A: $t$ (23)=6.073, p<0.001; AV: $t$ (23)=7.054, p<0.001; FDR corrected). In contrast, at gait-cycle frequency (1 Hz), only the response to AV stimulation showed significant peaks (V: $t$ (23)=–2.072, p=0.975; A: $t$ (23)=–0.054, p=0.521; AV: $t$ (23)=4.059, p<0.001; FDR corrected). Besides, we also observed significant peaks at 4 Hz in all three conditions (ps<0.001, FDR corrected), which showed a similar audiovisual integration mode as 2 Hz (see more details in Appendix and *Figure 2—figure supplement 1*).

Furthermore, we directly compared the cortical tracking effects between different conditions via a two-tailed paired t-test. At both 1 Hz (*Figure 2b*) and 2 Hz (*Figure 2c*), the amplitude in the AV condition was greater than that in the V condition (1 Hz: $t$ (23)=4.664, p<0.001, Cohen's $d$=0.952; 2 Hz: $t$ (23)=5.132, p<0.001, Cohen's $d$=1.048) and the A condition (1 Hz: $t$ (23)=2.391, p=0.025, Cohen's $d$=0.488; 2 Hz: $t$ (23)=3.808, p<0.001, Cohen's $d$=0.777), respectively, suggesting multisensory gains. More importantly, at 1 Hz, the amplitude in the AV condition was significantly larger than the algebraic sum of those in the A and V conditions ($t$ (23)=3.028, p=0.006, Cohen's $d$=0.618), indicating a super-additive audiovisual integration effect. While at 2 Hz, the amplitude in the AV condition was comparable to the unisensory sum ($t$ (23)=–0.623, p=0.539, Cohen's $d$=–0.127), indicating additive audiovisual integration.

### Experiment 1b

To further test whether such cortical tracking effect can apply to stimuli with a different speed, Experiment 1b altered the frequencies of the gait cycle and the corresponding step cycle to 0.83 Hz and 1.67 Hz while adopting the same paradigm as Experiment 1 a. Consistent with Experiment 1 a, the frequency-domain analysis revealed significant cortical tracking of the audiovisual stimuli at the new

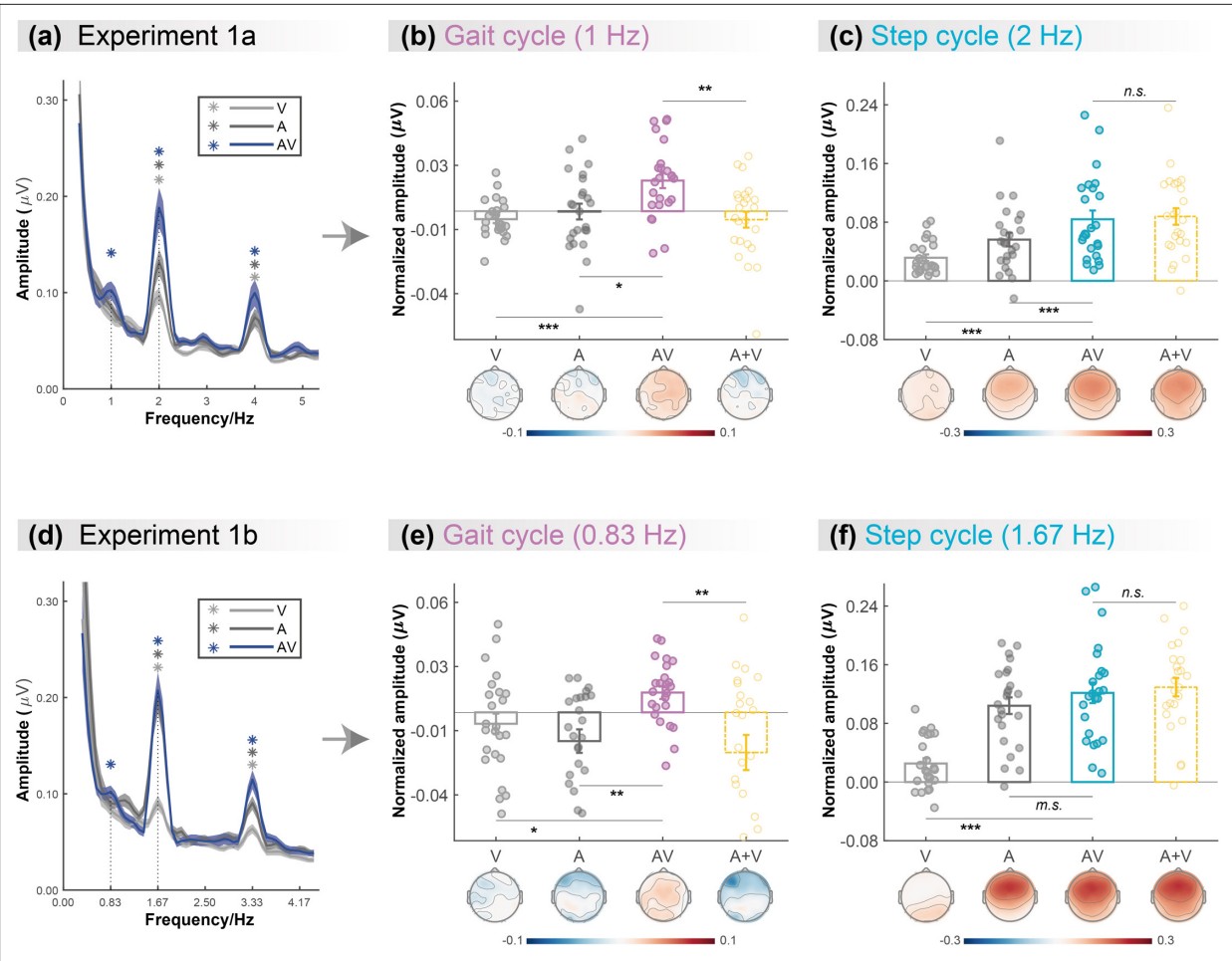

**Figure 2.** Cortical tracking of visual (V), auditory (A), and audiovisual (AV) biological motion (BM) signals at gait-cycle and step-cycle frequencies. (**a**) and (**d**) The amplitude spectra of electroencephalogram (EEG) responses in three conditions in Experiment 1 a and Experiment 1b, respectively. The solid lines show the grand average amplitude over all electrodes and subjects. The shaded regions depict standard errors of the group mean. Asterisks indicate significant spectra peaks (one-sample t-test against zero; p<0.05, FDR corrected). (**b**) and (**e**) The normalized amplitude at gait-cycle frequency in the AV condition exceeded the arithmetical sum of those in V and A conditions (AV >A + V, paired t-test), (**c**) and (**f**) but the normalized amplitude at step-cycle frequency in the AV condition was comparable to the sum of V and A (AV = A + V, paired t-test). Colored dots represent individual data in each condition. Error bars represent ±1 standard error of means (N = 24). *: p<0.05; **: p<0.01; ***: p<0.001; m.s.: 0.05<p<0.10; n.s.: p>0.05.

The online version of this article includes the following figure supplement(s) for figure 2:

**Figure supplement 1.** Cortical tracking of audiovisual biological motion (BM) information at different frequencies.

speeds. As shown in *Figure 2d*, both the responses to V, A, and AV stimuli showed clear peaks at step-cycle frequency (1.67 Hz; V: $t$ (23)=3.473, p=0.001; A: $t$ (23)=9.194, p<0.001; AV: $t$ (23)=8.756, p<0.001; FDR corrected) and its harmonics (3.33 Hz, ps <0.001, FDR corrected). In contrast, at gait-cycle frequency (0.83 Hz), only the response to AV stimuli showed significant peaks (V: $t$ (23)=−1.125, p=0.846; A: $t$ (23)=−2.449, p=0.989; AV: $t$ (23)=3.052, p=0.003; FDR corrected).

At both 0.83 Hz (*Figure 2e*) and 1.67 Hz (*Figure 2f*), the amplitude in the AV condition was stronger or marginally stronger than that in the V condition (0.83 Hz: $t$ (23)=2.665, p=0.014, Cohen's $d$=0.544; 1.67 Hz: $t$ (23)=6.380, p<0.001, Cohen's $d$=1.302) and the A condition (0.83 Hz: $t$ (23)=3.625, p<0.001, Cohen's $d$=0.740; 1.67 Hz: $t$ (23)=1.752, p=0.093, Cohen's $d$=0.358), respectively, suggesting multisensory gains. More importantly, at 0.83 Hz, the amplitude in the AV condition was significantly larger than the sum of those in the A and V conditions ($t$ (23)=3.240, p=0.004, Cohen's $d$=0.661), indicating a super-additive audiovisual integration effect. By contrast, at 1.67 Hz, the amplitude in the AV condition was comparable to the unisensory sum ($t$ (23)=−0.735, p=0.470, Cohen's $d$=−0.150), indicating linear audiovisual integration. Significant peaks were also observed at 3.33 Hz in all three conditions

(ps<0.001, FDR corrected), which showed similar audiovisual integration mode as 1.67 Hz (see more details in Appendix and *Figure 2—figure supplement 1*).

In summary, results from Experiments 1 a and 1b consistently showed that the cortical tracking of the audiovisual signals at different temporal scales exhibit distinct audiovisual integration modes, i.e., the super-additive effect at gait-cycle frequency and the additive effect at step-cycle frequency, indicating that the cortical tracking effects at the two temporal scales might be driven by functionally dissociable mechanisms.

## Cortical tracking of higher-order rhythmic structure contributes to the AVI of BM

To further explore whether and how the cortical tracking of rhythmic structures contributes to the specialized audiovisual processing of BM, we adopted both upright and inverted BM stimuli in Experiment 2. The task and the frequencies of visual stimuli in Experiment 2 were same as Experiment 1 a. Specifically, participants were required to perform the change detection task when perceiving upright and inverted visual BM sequences (1 Hz for gait-cycle frequency and 2 Hz for step-cycle frequency)

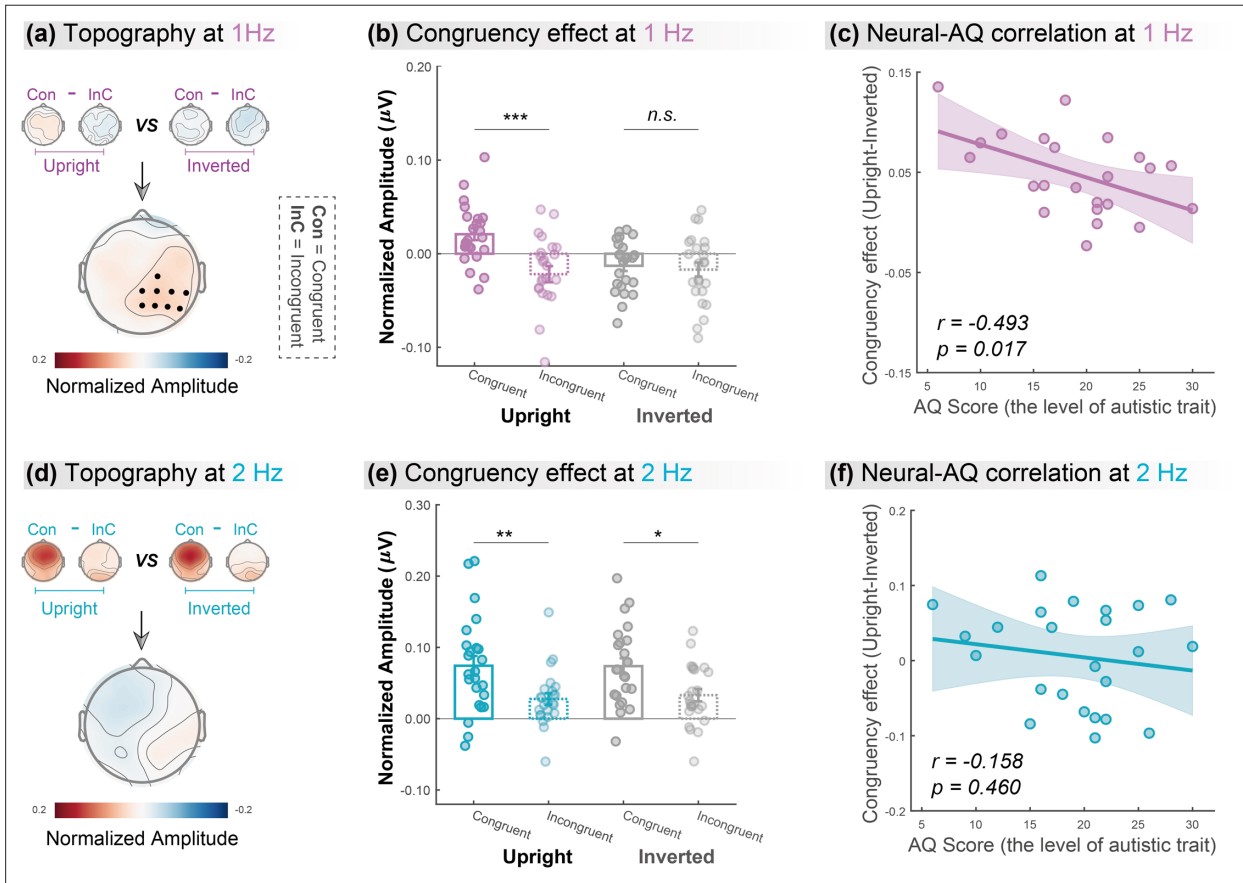

**Figure 3.** Cortical tracking at gait-cycle rather than step-cycle frequency contributes to the biological motion (BM)-specific audiovisual integration (AVI) effect. The lower panels in (**a**) and (**d**) depict the topographic maps of the BM-specific AVI effect, measured by the difference of congruency effects between the upright and inverted conditions at 1 Hz and 2 Hz, respectively. A significantly enhanced congruency effect in the upright condition relative to the inverted condition was observed at 1 Hz (marked by black dots) but not at 2 Hz (cluster-based permutation test; n = 1000, alpha = 0.05). The amplitude at these significant electrodes was averaged to quantify the congruency effect for the upright and inverted conditions at 1 Hz (**b**) and 2 Hz (**e**). Error bars represent ±1 standard error of means. N = 24. Paired t-test. *: p<0.05; **: p<0.01; ***: p<0.001; n.s.: p>0.05. Individuals' autistic traits correlated with the BM-specific AVI effect at 1 Hz (**c**) but not at 2 Hz (**f**). Shaded regions indicate the 95% confidence intervals.

The online version of this article includes the following figure supplement(s) for figure 3:

**Figure supplement 1.** Control analysis at step-cycle frequency.

**Figure supplement 2.** The biological motion (BM)-specific cortical tracking effect for high and low Autism-Spectrum Quotient (AQ) groups at 1 Hz (N = 23) and 2 Hz (N = 24).

accompanied by frequency congruent (1 Hz) or incongruent (0.6 Hz and 1.4 Hz) footstep sounds (*Figure 1c*). The audiovisual congruency effect, characterized by stronger neural responses in the audiovisual congruent condition compared with the incongruent condition, can be taken as an index of AVI (*Fleming et al., 2020*; *Maddox et al., 2015*; *Wuerger et al., 2012a*). A stronger congruency effect in the upright condition relative to the inverted condition characterizes an AVI process specific to BM information.

We contrasted the congruency effect between the upright and inverted conditions to search for clusters showing a significant difference, which equaled identifying an interaction effect, using a cluster-based permutation test over all electrodes (n=1000, alpha = 0.05; see Materials and methods). At 1 Hz, the congruency effect in the upright condition was significantly stronger than that in the inverted condition in a cluster at the right hemisphere (*Figure 3a*, lower panel, p=0.029; C2, CPz, CP2, CP4, CP6, Pz, P2, P4, P6). Then, we averaged the amplitude of electrodes within the significant cluster and performed two-tailed paired t-tests to examine whether the congruency effect was significant in the upright and the inverted conditions, respectively. Results showed that (*Figure 3b*) audiovisual congruency enhanced the oscillatory amplitude only for upright BM ($t$ (23)=4.632, p<0.001, Cohen's $d$=0.945) but not when visual BM was inverted ($t$ (23)=0.480, p=0.635, Cohen's $d$=0.098). Together, these findings suggest that cortical tracking of the high-order gait cycles involves a domain-specific process exclusively engaged in the AVI of BM.

In contrast, at 2 Hz, no cluster showed a significantly different congruency effect between the upright and inverted conditions (*Figure 3d*). We then conducted further analysis based on the electrodes yielded by 1 Hz as marked in *Figure 3a*. Results showed that both upright and inverted stimuli induced a significant congruency effect at 2 Hz (*Figure 3e*; upright: $t$ (23)=3.096, p=0.005, Cohen's $d$=0.632; inverted: $t$ (23)=2.672, p=0.014, Cohen's $d$=0.545). These findings suggest that neural tracking of the lower-order step cycles is associated with a domain-general AVI process mostly driven by temporal correspondence in physical stimuli.

## BM-specific cortical tracking correlates with autistic traits

Furthermore, we examined the link between individuals' autistic traits and the neural responses underpinning the AVI of BM, measured by the difference of congruency effect between the upright and the inverted BM conditions, using Pearson correlation analysis. After removing one outlier (whose neural response exceeded 3 SD from the group mean), we observed an evident negative correlation between individuals' AQ scores and their neural responses at 1 Hz (*Figure 3c*, r=–0.493, p=0.017) but not at 2 Hz (*Figure 3f*, r=–0.158, p=0.460). The lack of significant results at 2 Hz was not attributable to electrode selection bias based on the significant cluster at 1 Hz, as similar results were observed when we performed analyses on clusters showing non-selective significant congruency effects at 2 Hz (see the control analysis in Appendix and *Figure 3—figure supplement 1*). Besides, we split the participants based on their median AQ score and found that, compared with the high AQ group, the low AQ group showed a greater BM-specific cortical tracking effect at 1 Hz but not at 2 Hz. These findings provide further support to the possible linkage between social cognition and cortical tracking of BM as well as its dissociation at the two temporal scales (see more details in Appendix and *Figure 3—figure supplement 2*).

## Discussion

The current study investigated the neural implementation for the AVI of human BM information. We found that, even under a motion-irrelevant color detection task, observers' neural activity tracked the temporally corresponding audiovisual BM signals at the frequencies of two rhythmic structures, i.e., the higher-order structure of gait cycle at a larger integration window and the basic-level structure of step cycle at a smaller integration window. The strength of these cortical tracking effects was enhanced under the audiovisual condition than in the visual-only or auditory-only condition, indicating multisensory gains. More crucially, although the cortical tracking of both gait-cycle and step-cycle gain benefits from multisensory correspondence, the mechanisms underlying these two processes appear to be different. At step-cycle frequency, the cortical tracking effect in the AV condition equaled the additive sum of the unisensory conditions. Such linear integration may result from concurrent, independent processing of unisensory inputs without additional interaction of them (*Stein et al., 2009*).

In contrast, at gait-cycle frequency, the congruent audiovisual signals led to a super-additive multisensory enhancement over the linear combination of auditory and visual conditions (AV >A + V), despite that there was no evident cortical tracking effect in the visual condition, different from previous findings obtained with a motion-relevant change detection task (*Shen et al., 2023b*). This super-additive multisensory enhancement may bring about decreased thresholds of detection and identification (*Stanford et al., 2005*), allowing people to achieve a more clear and stable perception of the external environment and detect weak stimulus changes in time and respond adaptively.

Furthermore, results from Experiment 2 demonstrated that the cortical tracking of rhythmic structure corresponding to the gait cycle rather than the step cycle underlies the specialized processing of audiovisual BM information. In particular, the AVI effect at step-cycle frequency was significant for both upright and inverted BM signals and comparable between the two conditions, while the AVI effect at gait-cycle frequency was only significant in the upright condition and was greater than that in the inverted condition. The inversion effect has long been regarded as a marker of the specificity of BM processing in numerous behavioral and neuroimaging studies (*Grossman and Blake, 2001*; *Ma et al., 2022*; *Shen et al., 2023b*; *Simion et al., 2008*; *Troje and Westhoff, 2006*; *Vallortigara and Regolin, 2006*; *Wang et al., 2014*; *Wang and Jiang, 2012*; *Wang et al., 2022*). Our current findings of the inversion effect in the cortical tracking of audiovisual BM at the gait-cycle frequency suggest that the neural encoding of the higher-order rhythmic structure reflects the AVI of BM and contributes to the specialized processing of BM information. In contrast, the cortical tracking of the step cycle may reflect the integration of basic motion signals and corresponding sounds. Together, these results reveal that the neural tracking of rhythmic structures at different temporal scales plays distinct roles in the AVI of BM, which may result from the interplay of stimulus-driven and domain-specific mechanisms.

Besides the temporal dynamics of neural activity revealed by the cortical tracking process, we found that the BM-specific AVI effect was associated with neural activity in the right temporoparietal electrodes. This finding likely relates to the activation of the right posterior superior temporal sulcus (pSTS), a region responding to both auditory and visual BM information and being causally involved in BM perception (*Bidet-Caulet et al., 2005*; *Grossman et al., 2005*; *Wang et al., 2022*). While previous fMRI studies have observed STS activation when processing spatial or semantic correspondence between audiovisual BM (*Meyer et al., 2011*; *Wuerger et al., 2012b*), whether this region also engages in the audiovisual processing of BM signals based on temporal correspondence remains unknown. The current study provides preliminary evidence for such a possibility, inviting future research to localize the exact source of the multisensory integration processes based on imaging data with high spatial and temporal resolutions, such as MEG.

Cortical tracking of external rhythms is also described as cortical entrainment in a broad sense (*Ding et al., 2016*; *Obleser and Kayser, 2019*). Controversy remains regarding the involvements of endogenous neural oscillations and stimulus-evoked responses in these processes (*Duecker et al., 2024*), as it is challenging to fully dissociate these components due to their intricate interplay (*Herrmann et al., 2016*; *Hosseinian et al., 2021*). Despite the complexity of the neuronal mechanisms, previous research suggests that cortical tracking or entrainment plays a role in the multisensory processing of simple or discrete rhythmic signals (*Covic et al., 2017*; *Keitel and Müller, 2016*; *Nozaradan et al., 2012b*). These findings may partially explain the non-selective AVI effect at the step cycle in the current study. However, we found that the cortical tracking of the higher-order rhythmic structure formed by spatiotemporal integration of meaningful BM information (i.e. the gait cycle of upright walkers rather than inverted walkers) is selectively engaged in the AVI of BM, suggesting that the multisensory processing of natural continuous stimuli may involve unique mechanisms besides the purely stimulus-driven AVI process. These findings advance our understanding of the recently proposed view that multi-timescale neural processes coordinate multisensory integration (*Senkowski and Engel, 2024*), especially from the perspective of natural stimuli processing. Similar to BM, other natural rhythmic stimuli, like speech and music, also convey hierarchical structures that can entrain neural oscillations at different temporal scales, both in unisensory (*Ding et al., 2016*; *Doelling and Poeppel, 2015*) and multisensory contexts (*Biau et al., 2022*; *Crosse et al., 2015*; *Nozaradan et al., 2016*). Possibly, the audiovisual processing of these stimuli engages multiscale neural coding mechanisms that play distinct functions in perception. Investigating this issue and comparing the results with BM studies will help complete the picture of how the human brain

integrates complex, rhythmic information sampled from different sensory modalities to orchestrate perception in natural scenarios.

Since the scope of the current study mainly focused on the neural processing of BM, the task we employed was unrelated to audiovisual correspondence and not for establishing a direct link between neural responses and behavior. This is a limitation of the current study. A recent study demonstrated that listening to frequency-congruent footstep sounds, compared with incongruent sounds, enhanced the visual search for human walkers but not for non-BM stimuli containing the same rhythmic signals, indicating that audiovisual correspondence specifically enhances the perceptual and attentional processing of BM (*Shen et al., 2023a*). Future research could examine whether the cortical tracking of rhythmic structures plays a functional role in such behaviorally relevant tasks. They could also apply advanced neuromodulation techniques to elucidate the causal relevance of the cortical tracking effect to BM perception (e.g. *Kösem et al., 2020*; *Kösem et al., 2018*).

Last but not least, our study demonstrated that the selective cortical tracking of higher-level rhythmic structure in audiovisually congruent BM signals negatively correlated with individual autistic traits. This finding highlights the critical role of the neural tracking of audiovisual BM signals in social cognition. It also offers the first evidence that differences in audiovisual BM processing are already present in nonclinical individuals at the neural level and associated with their autistic traits, extending previous behavioral evidence for atypical audiovisual BM processing in ASD populations (*Falck-Ytter et al., 2013*; *Klin et al., 2009*). Meanwhile, given that impaired audiovisual BM processing at the early stage may influence social development and result in cascading consequences for lifetime impairments in social interaction (*Falck-Ytter et al., 2018*; *Klin et al., 2005*), it is worth exploring neural tracking of audiovisual BM signals in children, which may pave the way for utilizing it as a potential early neural marker for ASD.

## Materials and methods
### Participants
Seventy-two participants (mean age ± SD = 22.4±2.6 years, 35 females) took part in the study, 24 for each of Experiment 1 a, Experiment 1b, and Experiment 2. All of them had normal or corrected-to-normal vision and reported no history of neurological, psychiatric, or hearing disorders. They were naïve to the purpose of the study and gave informed consent according to procedures and protocols approved by the institutional review board of the Institute of Psychology, Chinese Academy of Sciences (reference number for approval: H21041).

### Stimuli
#### Visual stimuli
The visual stimuli (*Figure 1a*, left panel) consisted of 13 point-light dots attached to the head and major joints of a human walker (*Vanrie and Verfaillie, 2004*). The point-light walker was presented at the center of the screen without translational motion. It conveys rhythmic structures specified by recurrent forward motions of bilateral limbs (*Figure 1a*, right panel). Each step, regardless of left or right foot, occurs recurrently to form a step cycle. The antiphase oscillations of limbs during two steps characterize a gait cycle (*Shen et al., 2023b*). In Experiment 1 a, a full gait cycle took 1 s and was repeated six times to form a 6 s walking sequence. That is, the gait-cycle frequency is 1 Hz and the step-cycle frequency is 2 Hz. In Experiment 1b, the gait-cycle frequency was 0.83 Hz and the step-cycle frequency was 1.67 Hz. The gait cycle was repeated six times to form a 7.2 s walking sequence. The stimuli in Experiment 2 were the same as that in Experiment 1 a. Meanwhile, the point-light BM was flipped vertically to generate inverted BM (*Figure 1a*, left panel), which preserves the temporal structure of the stimuli but distorts its distinctive kinematic features, such as movement that is compatible with the effect of gravity (*Shen et al., 2023b*; *Troje and Westhoff, 2006*; *Wang et al., 2022*).

#### Auditory stimuli
Auditory stimuli were continuous footstep sounds (6 s) with a sampling rate of 44,100 Hz. As shown in *Figure 1b*, in Experiments 1 a and 2, the gait-cycle frequency of congruent sounds was 1 Hz, which consisted of two steps or two impulses generated by each foot striking the ground within one gait cycle. The incongruent sounds included a faster (1.4 Hz) and a slower (0.60 Hz) sound. Both congruent

and incongruent sounds were generated by manipulating the temporal interval between two successive impulses based on the same auditory stimuli. In Experiment 1b, the gait-cycle frequency of sound was 0.83 Hz.

## Stimuli presentation

The visual stimuli were rendered white against a gray background and displayed on a CRT (cathode ray tube) monitor. Participants sat 60 cm from the computer screen (1280×1024 at 60 Hz; high: 37.5 cm; width: 30 cm), with their heads held stationary on a chinrest. The auditory stimuli were presented binaurally over insert earphones. All stimuli were generated and presented using MATLAB together with the Psychophysics Toolbox (*Brainard, 1997*; *Pelli, 1997*).

## Procedure and task

### Experiment 1a

The experiment was conducted in an acoustically dampened and electromagnetically shielded chamber. Participants completed the task under three conditions (visual: V; auditory: A; audiovisual: AV) with the same procedure (*Figure 1c*) except for the stimuli. In the V condition, each trial began with a white fixation cross (0.42°×0.42°) displayed at the center of a gray background for a random duration (0.8–1 s). Subsequently, a 6 s point-light walker (3.05°×5.47°) walked toward the left or right at a constant walking cycle frequency (1 Hz). To maintain observers' attention, 17–23% of the trials were randomly selected as catch trials, in which the color of the walker changed (the RGB values changed from [255 255 255] to [207 207 207]) one or two times throughout the trial. Each change lasted 0.5 s. Observers were required to report the number of changes (0, 1, or 2) via keypresses as accurately as possible after the point-light display was replaced by a red fixation. The next trial started 2–3 s after the response. In the A condition, the 6 s stimuli were replaced by a visually static BM figure accompanied by continuous footstep sounds. The frequency of footstep sounds was congruent with the frequency of visual BM in the V condition. In the AV condition, the stimuli were temporally congruent visual BM sequences (as in the V condition) and footstep sounds (as in the A condition). Three conditions were conducted in separate blocks. V condition was performed in the middle of A and AV conditions. The order of A and AV conditions was counterbalanced across participants. Each participant completed 40 experimental trials without changes and 10–15 catch trials in each condition, resulting in a total of 150–165 trials. In each condition, participants completed a practice session with three trials to get familiar with the task before the formal EEG experiment.

### Experiment 1b

The procedure of Experiment 1b was the same as that for Experiment 1 a but with two exceptions. First, to test if the cortical tracking effect can apply to stimuli with a different speed, we altered the frequencies of gait and step cycles to 0.83 Hz and 1.67 Hz. Second, we presented the three conditions (V, A, and AV) in a completely random order to eliminate the influence of presentation order. To minimize the potential influence of condition switch, we increased the trial number in the practice session from 3 to 14 for each condition.

### Experiment 2

The procedure in Experiment 2 was similar to the AV condition in Experiment 1 a, except that the visually displayed BM was accompanied by frequency congruent (1 Hz) or incongruent (0.6 Hz or 1.4 Hz) footstep sounds. Each participant completed a total of 76 experiment trials, consisting of 36 congruent-trials, 20 incongruent-trials with faster sounds (1.4 Hz), and 20 incongruent-trials with slower sounds (0.6 Hz). These trials were assigned to three blocks based on the frequency of the footstep sounds, with the order of the three frequencies balanced across participants. Besides, an inverted BM was used as a control to investigate whether there is a specialized mechanism tuned to the AVI of life motion signals. The order of upright and inverted conditions was balanced across participants. Meanwhile, we measured the participants' autistic traits by using the Autism-Spectrum Quotient, or AQ questionnaire (*Baron-Cohen et al., 2001*). Higher AQ scores indicate a higher level of autistic traits.

## EEG recording and analysis

EEG was recorded at 1000 Hz using a SynAmps[2] NeuroScan amplifier System with 64 electrodes placed on the scalp according to the international 10–20 system. Horizontal and vertical eye movements were measured via four additional electrodes placed on the outer canthus of each eye and the inferior and superior areas of the left orbit. Impedances were kept below 5 kΩ for all electrodes.

### Preprocessing

The catch trials were excluded from EEG analysis. All preprocessing and further analyses were performed using the FieldTrip toolbox (*Oostenveld et al., 2011*; http://fieldtriptoolbox.org) in the MATLAB environment. EEG recordings were pass-filtered between 0.1 Hz and 30 Hz, and down-sampled to 100 Hz. Then the continuous EEG data were cut into epochs ranging from –1 s to 6 gait cycles (7.2 s in Experiment 1b and 6 s in other experiments) time-locked to the onset of the visual point-light stimuli. The epochs were visually inspected, and trials contaminated with excessive noise were excluded from the analysis. After the trial rejection, eye and cardiac artifacts were removed via independent component analysis based on the Runica algorithm (*Bell and Sejnowski, 1995*; *Jung et al., 2000*; *Makeig, 2002*). Then the cleaned data were re-referenced to the average mastoids (M1 and M2). To minimize the influence of stimulus-onset evoked activity on EEG spectral decomposition, the EEG recording before the onset of the stimulus and the first cycle (1 s in Experiments 1 a and 2; 1.2 s in Experiment 1b) of each trial was excluded (*Nozaradan et al., 2012a*). After that, the EEG epochs were averaged across trials for each participant and condition.

### Frequency-domain analysis and statistics

A fast Fourier transform (FFT) with zero padding (1200) was used to convert the averaged EEG signals from the temporal domain to the spectral domain, resulting in a frequency resolution of 0.083 Hz, i.e., 1/12 Hz, which is sufficient for observing neural responses around the frequency of the rhythmic BM structures in all experiments. When performing FFT, a Hanning window was adopted to minimize spectral leakage. Then, to remove the 1 /f trend of the response amplitude spectrum and identify spectral peaks, the response amplitude at each frequency was normalized by subtracting the average amplitude measured at the neighboring frequency bins (two bins on each side) (*Nozaradan et al., 2012a*). We calculated the normalized amplitude separately for each electrode (except for electrooculogram electrodes, CB1, and CB2), participant, and condition.

In Experiment 1, the normalized amplitude in all electrodes was averaged and a right-tailed one-sample t-test against zero was performed on the grand average amplitude to test whether the neural response in each frequency bin showed a significant tracking effect or spectral peak. This test was applied to all frequency bins below 5.33 Hz and multiple comparisons were controlled by false discovery rate (FDR) correction at $p<0.05$ (*Benjamini and Hochberg, 1995*). In Experiment 2, to further identify the BM-specific AVI process, the audiovisual congruency effect was compared between the upright and inverted conditions using a cluster-based permutation test over all electrodes (1000 iterations, requiring a cluster size of at least two significant neighbors, a two-sided t-test at $p<0.05$ on the clustered data) (*Oostenveld et al., 2011*; http://fieldtriptoolbox.org). This allowed us to identify the spatial distribution of the BM-specific congruency effect.

# Additional information

### Funding

| Funder | Grant reference number | Author |
| --- | --- | --- |
| Ministry of Science and Technology of the People's Republic of China | 2021ZD0204200 | Ying Wang |
| Ministry of Science and Technology of the People's Republic of China | 2021ZD0203800 | Yi Jiang |

| Funder | Grant reference number | Author |
|---|---|---|
| National Natural Science Foundation of China | 32171059 | Ying Wang |
| National Natural Science Foundation of China | 32430043 | Yi Jiang |
| Youth Innovation Promotion Association of the Chinese Academy of Sciences | | Ying Wang |
| China Postdoctoral Science Foundation | 2024M170993 | Li Shen |
| China Postdoctoral Science Foundation | 2024M753476 | Li Shen |
| The Key Research and Development Program of Guangdong, China | 2023B0303010004 | Yi Jiang |
| Interdisciplinary Innovation Team of the Chinese Academy of Sciences | JCTD-2021-06 | Yi Jiang |
| Fundamental Research Funds for the Central Universities | | Yi Jiang |

The funders had no role in study design, data collection and interpretation, or the decision to submit the work for publication.

## Author contributions

Li Shen, Conceptualization, Data curation, Formal analysis, Funding acquisition, Investigation, Visualization, Methodology, Writing – original draft, Writing – review and editing; Shuo Li, Yuhao Tian, Investigation, Writing – original draft; Ying Wang, Conceptualization, Supervision, Funding acquisition, Methodology, Writing – review and editing; Yi Jiang, Conceptualization, Supervision, Writing – review and editing, Funding acquisition

## Author ORCIDs

Li Shen ⑩ https://orcid.org/0000-0002-6088-7892
Ying Wang ⑩ https://orcid.org/0000-0002-5756-2480
Yi Jiang ⑩ https://orcid.org/0000-0002-5746-7301

## Ethics

All procedures contributing to this work comply with the ethical standards of the relevant national and institutional committees on human experimentation and with the Helsinki Declaration of 1975, as revised in 2008. Written informed consent, and consent to publish, was obtained from participants. The institutional review board of the Institute of Psychology, Chinese Academy of Sciences has approved this study (reference number for approval: H21041).

Reviewer #1 (Public review): https://doi.org/10.7554/eLife.98701.5.sa1
Reviewer #2 (Public review): https://doi.org/10.7554/eLife.98701.5.sa2
Author response https://doi.org/10.7554/eLife.98701.5.sa3

# Additional files

## Supplementary files

MDAR checklist

## Data availability

The data and code accompanying this study are made available at https://doi.org/10.57760/sciencedb.psych.00144.

The following dataset was generated:

| Author(s) | Year | Dataset title | Dataset URL | Database and Identifier |
|---|---|---|---|---|
| Shen L | 2023 | Data from: Cortical tracking of hierarchical rhythms orchestrates the multisensory processing of biological motion | https://doi.org/10.57760/sciencedb.psych.00144 | Science Data Bank, 10.57760/sciencedb.psych.00144 |

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

## Appendix

### Results on other peaks in Experiment 1

As shown in *Figure 2a and d*, the audiovisual BM signals induced significant amplitude peaks at 1 f (1/0.83 Hz), 2 f (2/1.67 Hz), and 4 f (4/3.33 Hz) relative to the gait cycle frequency (ps <0.001; FDR corrected). To further test the roles of the neural activity at different frequencies, we analyzed the AVI modes at each frequency, by comparing the neural responses in the AV condition with the sum of those in the A and V conditions. Given that Experiments 1 a and 1b yielded similar results, we collapsed the data and presented the results as follows.

As shown in *Figure 2—figure supplement 1*, at 4 f, the amplitude of neural responses showed significant peaks in all three conditions (V: *t* (47)=6.869, p<0.001; A: *t* (47)=7.938, p<0.001; AV: *t* (47)=8.303, p<0.001). Moreover, the amplitude in the AV condition was larger than that in the V condition (*t* (47)=4.855, p<0.001, Cohen's *d*=0.701) and the A condition (*t* (47)=3.080, p=0.003, Cohen's *d*=0.445), respectively, suggesting multisensory gains. In addition, the amplitude in the AV condition was comparable to the unisensory sum (*t* (47)=–1.049, p=0.300, Cohen's *d*=–0.151), indicating linear AVI. These results were similar to those observed at 2 f but different from those at 1 f, as reported in the main text. Together, these results show a similar additive AVI mode at 2 f and 4 f and a super-additive integration mode only at 1 f, suggesting that the cortical tracking effects at 2 f and 4 f may be functionally linked but independent of that at 1 f.

### Control analysis of correlation in Experiment 2

The control analysis mainly aims to eliminate the potential bias due to electrode selection. As reported in the main text, both correlation analyses at 1 Hz and 2 Hz were performed based on electrodes in the significant cluster observed at 1 Hz because there was no significant cluster at 2 Hz (*Figure 3a and d*, lower panel). There is a possibility that these electrodes did not show a significant congruency effect at 2 Hz, either in the upright or the inverted condition, thus were not able to capture the correlation between the variance in neural responses and that in autistic traits. To rule out such a possibility, we conducted a control analysis based on electrodes showing a significant congruency effect at 2 Hz, for the upright (p=0.004, cluster-based permutation test) and inverted (p=0.002, cluster-based permutation test) conditions, respectively. As shown in *Figure 3—figure supplement 1a*, the difference of congruency effect between upright and inverted conditions is still not significant in the group level (*t* (23)=–0.689, p=498, Cohen's *d*=–0.141), while it shows individual variance (SD = 0.079, range: [–0.173 0.153]) larger than that for the 1 Hz condition (SD = 0.041, range: [–0.023 0.135]), which allows us to identify a correlation if existing. Analysis of these data showed a non-significant correlation (*Figure 3—figure supplement 1b*, *r*=–0.091, p=0.674), similar to the results illustrated in *Figure 3f*.

### Additional analysis in Experiment 2

To further examine the linkage between autistic traits and the BM-specific cortical tracking effect, we split the participants into high (above 20) and low (below or equal to 20) AQ groups by the median AQ score (20) of this sample. Similar to correlation analysis, one outlier, whose BM-specific audiovisual congruency effect (upright – inverted) in neural responses at 1 Hz exceeds 3 SD from the group mean, was removed from the following analysis. As shown in *Figure 3—figure supplement 2*, at 1 Hz, participants with low AQ showed a greater cortical tracking effect compared with high AQ participants (*t* (21)=2.127, p=0.045). At 2 Hz, low and high AQ participants showed comparable neural responses (*t* (22)=0.946, p=0.354). These results are in line with the correlation analysis, providing further support to the relevance between social cognition and cortical tracking of BM as well as its dissociation at the two temporal scales.

