## [Editor Report · eLife Assessment]

Wang et al. presented visual (dot) motion and/or the sound of a walking person and found **solid** evidence that EEG activity tracks the step rhythm, as well as the gait (2-step cycle) rhythm, with some demonstration that the gait rhythm is tracked superadditively (power for A+V condition is higher than the sum of the A-only and V-only condition). The **valuable** findings will be of wide interest to those examining biological motion perception and oscillatory processes more broadly.

---

## [Referee Report · Reviewer #1 (Public review)]

Shen et al. conducted three experiments to study the cortical tracking of the natural rhythms involved in biological motion (BM), and whether these involve audiovisual integration (AVI). They presented participants with visual (dot) motion and/or the sound of a walking person. They found that EEG activity tracks the step rhythm, as well as the gait (2-step cycle) rhythm. The gait rhythm specifically is tracked superadditively (power for A+V condition is higher than the sum of the A-only and V-only condition, Experiments 1a/b), which is independent of the specific step frequency (Experiment 1b). Furthermore, audiovisual integration during tracking of gait was specific to BM, as it was absent (that is, the audiovisual congruency effect) when the walking dot motion was vertically inverted (Experiment 2). Finally, the study shows that an individual's autistic traits are negatively correlated with the BM-AVI congruency effect.

---

## [Referee Report · Reviewer #2 (Public review)]

The authors evaluate spectral changes in electroencephalography (EEG) data as a function of the congruency of audio and visual information associated with biological motion (BM) or non-biological motion. The results show supra-additive power gains in the neural response to gait dynamics, with trials in which audio and visual information was presented simultaneously producing higher average amplitude than the combined average power for auditory and visual conditions alone. Further analyses suggest that such supra-additivity is specific to BM and emerges from temporoparietal areas. The authors also find that the BM-specific supra-additivity is negatively correlated with autism traits.

---

## [Author Response]

The following is the authors’ response to the previous reviews.

**Public Reviews:**

**Reviewer #1 (Public review):**
Summary:Shen et al. conducted three experiments to study the cortical tracking of the natural rhythms involved in biological motion (BM), and whether these involve audiovisual integration (AVI). They presented participants with visual (dot) motion and/or the sound of a walking person. They found that EEG activity tracks the step rhythm, as well as the gait (2-step cycle) rhythm. The gait rhythm specifically is tracked superadditively (power for A+V condition is higher than the sum of the A-only and V-only condition,Experiments 1a/b), which is independent of the specific step frequency (Experiment 1b). Furthermore, audiovisual integration during tracking of gait was specific to BM, as it was absent (that is, the audiovisual congruency effect) when the walking dot motion was vertically inverted (Experiment 2). Finally, the study shows that an individual's autistic traits are negatively correlated with the BM-AVI congruency effect.Strengths:The three experiments are well designed and the various conditions are well controlled. The rationale of the study is clear, and the manuscript is pleasant to read. The analysis choices are easy to follow, and mostly appropriate.Weaknesses:On revision, the authors are careful not to overinterpret an analysis where the statistical test is not independent from the data (channel) selection criterion.

Thanks for the suggestion and we have done this according to your recommendations below.

**Reviewer #1 (Recommendations for the authors):**
Re: the double-dipping concern: I appreciate the revision. Just to clarify: my concern rests with the selection of *electrodes* based on the interaction test for the 1Hz condition. The 2Hz condition analogous test yields no significant electrodes. You perform subsequent tests (t-tests and 3-way interaction) on the data averaged across the electrodes that were significant for the 1Hz condition. Therefore, these tests will be biased to find a pattern reflecting an interaction at 1Hz, while no similar bias exists for an effect at 2Hz. Therefore, there is a bias to observe a 3-way interaction, and simple effects compatible with a 2-way interaction only for 1Hz, not for 2Hz (which is exactly what you found). There is no good statistical alternative here, I appreciate that, but the bias exists nonetheless. I think the wording is improved in this revision, and the evidence is convincing even in light of this bias.

We are grateful for your thoughtful comments on the analytical methods. We appreciate your concerns regarding the potential bias of examining 3-way interaction based on electrodes yielding a 2-way interaction effect. To address this issue, we have conducted a bias-free analysis based on electrodes across the whole brain. The results showed a similar pattern of 3-way interaction as previously reported (p = 0.051), suggesting that the previous findings might not be caused by electrode selection. Given that the main results of Experiment 2 were not based on whole-brain analysis, we did not involve this analysis in the main text, and we have removed the three-way interaction results based on selected electrodes from the manuscript to reduce potential concerns. It is also noteworthy that, when performing analyses based on channels independent of the interaction effect at 1 Hz (i.e., significant congruency effects in the upright and inverted conditions, respectively, at 2Hz), we got similar results as reported in the main text (i.e., non-significant interaction and correlation at 2 Hz). These results were presented in the supplementary file in previous versions and mentioned in the correlation part of the Results section (see Fig. S2). Once again, we sincerely appreciate your careful review of our research. We hope the abovementioned points adequately address your concern.

**Reviewer #2 (Public review):**
Summary:The authors evaluate spectral changes in electroencephalography (EEG) data as a function of the congruency of audio and visual information associated with biological motion (BM) or non-biological motion. The results show supra-additive power gains in the neural response to gait dynamics, with trials in which audio and visual information was presented simultaneously producing higher average amplitude than the combined average power for auditory and visual conditions alone. Further analyses suggest that such supra-additivity is specific to BM and emerges from temporoparietal areas. The authors also find that the BM-specific supra-additivity is negatively correlated with autism traits.Strengths:The manuscript is well-written, with a concise and clear writing style. The visual presentation is largely clear. The study involves multiple experiments with different participant groups. Each experiment involves specific considered changes to the experimental paradigm that both replicate the previous experiment's finding yet extend it in a relevant manner.In the first revisions of the paper, the manuscript better relays the results and anticipates analyses, and this version adequately resolves some concerns I had about analysis details. In a further revision, it is clarified better how the results relate to the various competing hypotheses on how biological motion is processed.Weaknesses:Still, it is my view that the findings of the study are basic neural correlate results that offer only minimal constraint towards the question of how the brain realizes the integration of multisensory information in the service of biological motion perception, and the data do not address the causal relevance of observed neural effects towards behavior and cognition. The presence of an inversion effect suggests that the supraadditivity is related to cognition, but that leaves open whether any detected neural pattern is actually consequential for multi-sensory integration (i.e., correlation is not causation). In other words, the fact that frequency-specific neural responses to the [audio & visual] condition are stronger than those to [audio] and [visual] combined does not mean this has implications for behavioral performance. While the correlation to autism traits could suggest some relation to behavior and is interesting in its own right, this correlation is a highly indirect way of assessing behavioral relevance. It would be helpful to test the relevance of supra-additive cortical tracking on a behavioral task directly related to the processing of biological motion to justify the claim that inputs are being integrated in the service of behavior. Under either framework, cortical tracking or entrainment, the causal relevance of neural findings toward cognition is lacking.Overall, I believe this study finds neural correlates of biological motion that offer some constraint toward mechanism, and it is possible that the effects are behaviorally relevant, but based on the current task and associated analyses this has not been shown (or could not have been, given the paradigm).
**Reviewer #2 (Recommendations for the authors):**
Thank you for your revisions; I have updated the Strengths section, and reworded the weaknesses section. I now concede that the neural effects observed offer some constraint towards what the neural mechanisms for AV integration for BM are, whereas in my previous review, I said too strongly that these results do not offer any information about mechanism.

Thank you again for your insightful thoughts and comments on our research. They have contributed greatly to enhancing the discussion of the article and provided valuable inspiration for future exploration of causal mechanisms.